# Edible Coatings and Films for Preparation of Grapevine By-Product Infusions and in Freshly Processed Products

Teresa Pinto [1,*], Ana Pinto [2] and Alice Vilela [3]

1   CITAB, Centre for the Research and Technology of Agro-Environmental and Biological Sciences and Inov4Agro, Department of Biology and Environment, Institute for Innovation, Capacity Building and Sustainability of Agrifood Production, School of Life and Environmental Sciences, University of Trás-os-Montes and Alto Douro, 5000-801 Vila Real, Portugal
2   University of Trás-os-Montes and Alto Douro, 5000-801 Vila Real, Portugal
3   Chemistry Research Center (CQ-VR), Department of Agronomy, School of Agrarian and Veterinary Sciences, University of Trás-os-Montes and Alto Douro, 5000-801 Vila Real, Portugal
*   Correspondence: tpinto@utad.pt

**Abstract:** The wine industry is responsible for a considerable part of environmental problems because of the large amounts of residues. However, several studies have shown these wine industry residues, such as grapes, skins, seeds, and leaves, represent a complex matrix of bio-compounds, such as phenolic compounds, flavonoids, procyanidins, anthocyanins, tannins, catechin, quercetin, kaempferol, and trans-resveratrol, and nutrients such as vitamin C. These wine and vine by-products or their extracts have antioxidant, anti-inflammatory, cardioprotective, anti-aging, and anti-cancer activities, which benefit human health. Due to processing (drying, mincing), some vine by-products are perishable and may present a short shelf-life. The production of the developed products can be achieved by using edible films and coatings. The use of edible coatings is an innovative method for preservation in postharvest. This technique is gaining popularity since it is easy to apply, environmentally friendly, and highly efficient. Indeed, the application of edible layers on lightly processed foods can preserve their essential nutrients and protect them from attack by microorganisms in addition to preserving their appearance by maintaining their original color, flavor, and texture. More research must be carried out to optimize coating formulations to achieve the highest possible quality. This review aims to elucidate several techniques of making edible coatings and the different types of edible coatings that can be used in the preparation of grape by-products for foods and drinks, namely grapevine infusions made with dried minced grapes, dried minced grape pomaces, and in freshly processed products. Besides the usually used coating materials, such as chitosan, agar-agar, gelatin, and alginate, other compounds will also be discussed, namely guar gum, soy lecithin, maltodextrin, inulin, and propolis.

**Keywords:** casting; extrusion; chitosan; cellulose; pectin; carrageenan; alginate; gums

## 1. Introduction

According to the United Nations [1] in November 2022 the global human population reached 8.0 billion and is expected to increase to 9.7 billion in 2050. Feeding this population will be difficult, and alternatives to preserve food and natural resources must be found. The alternatives will be increasing production, expanding distribution, reducing postharvest losses, and reusing waste [2]. Waste is composed of all the products that are discarded in the food industry, for example, grapes that are left on the vine unpicked and pomace of some fruits like grapes, apples, pomegranates, tomatoes, carrots, and others [3–8].

Millions of tons of waste are produced during the winemaking process, and this waste is harmful to the environment [9]. Grapes and by-products of grapes, called grape pomace, are rich in bioactive compounds (lipids, dietary fibers, polyphenols), which make them valuable products. Grape is one of the most used and consumed fruits in the

world with a production of 75 million tons per year, of which 20 to 30% is considered waste, and only 3% of this residue is valued for the creation of feed and other purposes. Creating new food products, such as infusions, would be a solution to reduce waste and take advantage of the benefits that grape by-products give us [10]. Several studies have shown that pomace contains beneficial components for health, having several biological activities such as anti-atherosclerosis. They also present cardioprotective, antioxidant, neuroprotective, anti-inflammatory, antidiabetic, antiviral, and antimicrobial features that prevent the proliferation of cancer cells and may provide some protection against certain types of cancer [9,11].

Fresh fruits and vegetables typically have a natural, waxy coating and an edible coating that enhances their functionality. However, when it comes to fresh-cut products or minimally processed fruits, their surfaces are exposed to external environmental factors, such as microbes, which make them more susceptible to damage. Additionally, these products undergo various metabolic reactions, leading to changes in color, texture, and an accelerated ripening process [12].

The longer or shorter storage times of grape by-products depend on their exposure to the environment since changes in physical, chemical, and biological factors may cause their deterioration [13]. Microbial and oxidative deterioration is one of the main factors contributing to food waste, for it is necessary to create packaging or suitable protections for food while focusing on improving and preserving food quality. The creation of packaging contributes to the reduction of food waste and allows consumers to consume healthy, natural, and safe foods [14].

Currently, one of the alternatives for the prolongation of food shelf-life is the creation of plastic packaging, which is already implemented in the market and has been used in the daily lives of consumers for many years. The high usage of this alternative relies on its low cost and resistance, as well as being easy-to-mold [14,15]. Despite the aspects above-mentioned, there are several studies [16–18] that show that this solution is not a promising alternative, as it implies an overall increase in the use of plastic. In 2021, there were 390.700 million tonnes of plastic produced worldwide, of which 57.200 million tonnes were produced in Europe with a gradual increase over the years [19]. Another disadvantage of this material is the fact that it is not biodegradable, constituting a harmful alternative to the environment, something that is not beneficial for the lifestyle of the entire population and of all biodiversity. It is increasingly a more debatable and worrying issue, and because of this, several alternatives have been investigated for the substitution of this alternative for something both more sustainable and not harmful to the environment.

The alternative to replacing plastic and consequently meeting the requirements of the extension of the shelf-life of food is the creation of edible coatings that are progressively a more cost-effective, sustainable, and environmentally effective solution in the use of primary packaging in the food industry. They can also be an alternative to replace the commercial synthetic waxes that are made of oxidized polyethylene. These coatings protect food ingredients from oxidation and degradation caused by enzymes while also helping to preserve the flavors originating from each food, thus ensuring the viability of the active ingredients for a long time. Moreover, they may contain useful additives, like antioxidants and phytonutrients [2,20,21]. In addition, depending on the coating, properties that are beneficial to health may be added, such as preventing chronic degenerative and cardiovascular diseases due to phenolic compounds and preventing reproductive, nervous system, inflammation, and immune system degeneration [22,23].

Edible coatings are made from natural food-grade materials, such as hydrocolloids (polysaccharides, proteins), lipids, and emulsifiers, produced with different techniques, such as dipping (immersing), spraying, spreading, brushing, pressing them/thermoforming, or extrusion [24–26]. The most used technique for coating is immersing, where food is dipped in a liquid containing food matrices, forming a film around the food and protecting all the components present.

Food matrices can be originated from plants, animals, or microorganisms. Many compounds can be used in food coatings, such as chitosan, konjac glucomannan, carboxymethyl cellulose [27], pullulan, hidroxipropilmetil cellulose, alginate, seed gum (guar, locust bean, tara, basil, fenugreek), exudate gum (gum ghatti, Persian, tragacanth, shellac) [28], gum arabic, xanthan gum, maltodextrin, pectin, starch, dextran, milk proteins, cellulose, zein, galactomannan, whey protein, sodium caseinate, polydextrose [23], gelatin [15], propolis extracts [29], vanillin [30], inulin [20], soy lecithin, and carnauba wax [31], among others. Edible films with Chitosan are the most applied in the food industry, especially in food packaging, due to their good mechanical properties, selective permeability to $O_2$ and $CO_2$, and outstanding film-forming properties [32]. In previous studies, it was applied to different foods, such as grapes that preserve the quality for 6 days at 20 °C [33], chicken breast for 12 days at 4 °C [34], pork chunks for 12 days at 3 °C [35], and strawberries [36] and apple for 4 weeks [37]. Chitosan requires the incorporation of other components, such as different biopolymers, plasticizers, fillers, or antimicrobial components, to reduce its brittleness and improve the flexibility, toughness, and tear resistance of the material [38,39]. Chitosan with glycerol is one combination that was promising due to its ability to enhance film flexibility and improve mechanical properties [40]. In the study by Yu et al., [41] chitosan was mixed with different deep eutectic solvents (DES) and showed an improvement in the edible film depending on the DES. The addition of choline-based DES reduced water contact angles, decreased the decomposition temperature and thermal stability, and had strong UV shielding and antioxidant properties of the chitosan films [41]. In the study conducted by Díaz-Montes [42] and his colleagues, in 2021, progress was reported in the creation of sustainable films made from biopolymers, specifically chitosan and dextran. Dextran was obtained through biotechnological techniques using *Leuconostoc mesenteroides* SF3 (GenBank: KR362874). The resulting biofilms were carefully analyzed and tested for the preservation of mushrooms (*Agaricus bisporum*). The researchers concluded that the application of these films (though dispersions of 0.5% *w/v* dextran and 1% *w/v* chitosan) can be a viable alternative to commercial plastics since the biofilms allowed for extending the period of deterioration of the mushrooms by up to 28 days at 4 °C, maintaining the physical–chemical characteristics and visual appearance. Consequently, the authors suggest that based on this type of film produced, it is possible to direct its application to biodegradable packaging for fruits, vegetables, and minimally processed foods.

This review aims to describe the characteristics and usage of several polymer matrices, such as chitosan, agar–agar, gelatin, and alginate, as well as others, such as guar gum, soy lecithin, maltodextrin, inulin, and propolis. These compounds can be used in the preparation of food and beverages from grape by-products, namely infusions of minced dried grapes, pomace, and vine leaves.

## 2. Techniques of Making Edible Coatings

Grape and grape pomace is rich in compounds, like lipids, dietary fibers, minerals, and proteins, specifically catechins, gallic acid, and procyanidins [43]. Polyphenols are also one of the bioactive compounds that are formed during the production of wine. They can be found mostly in seeds, specifically phenolic acids, flavonoids, proanthocyanidins, and resveratrol [9]. The grape skin is rich in anthocyanins [9]. According to several studies, the bioactive compounds present in grape and grape pomace are beneficial to health and can be used in the production of drugs, food, and healthcare products [44–46]. Choleva et al. [47] showed that grape pomace extract helps in the reduction of cholesterol and blood glucose, improving insulin sensitivity. It is also able to modulate uric acid levels, lipid peroxidation, superoxide dismutase activity, and protein oxidation [47]. To preserve these compounds, alternatives must be explored, aiming to use the benefits of grape pomace for people's health, prolonging food shelf-life, and reducing food waste, specifically grape by-products.

In the food industry, there are several techniques aiming to preserve the quality of food products. Edible films and coatings are an example of this. However, there are differences

between both products. In edible films or edible packaging, the film involves the food and may or may not be removed before consumption, while the coating is used directly on the surface of the food and is consumed together with the food [23,48]. This solution, in the last few years, has been gradually accepted by consumers due to its low cost and the fact that the polymers are not toxic, do not pollute the environment, and retain the compounds in the food that are beneficial to human health [49]. These techniques provide a semi-permeable barrier that protects the food against undesirable compounds, such as gases and moisture. This barrier lowers respiration, water loss, and oxidation reaction [50].

For the manufacturing of films and coatings, different techniques will be presented in this review (Figure 1), as we will also talk about the different matrices that can be used in the realization of edible coatings.

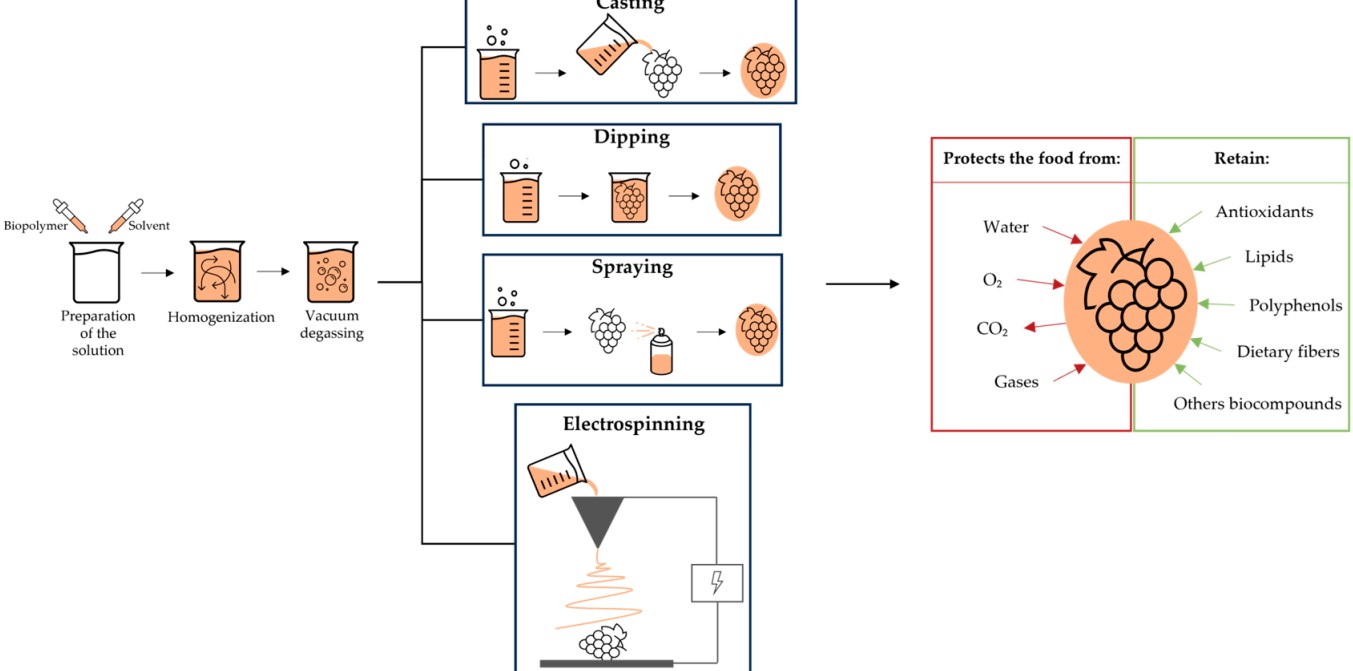

**Figure 1.** Illustration of the application and protection of edible coatings in grapevine by-products.

The selection of the application method depends on the characteristics of the surface and the geometry of the product to be coated. Regarding this, it can be stated that physical properties such as surface tension and plasticity modify the mechanical properties of edible coatings [51].

### 2.1. Casting and Extrusion

For the making of edible films, two techniques can be used, namely casting and extrusion (Figure 1). Casting is a technique used in the laboratory that has the advantage of being easy and simple. This technique is divided into three phases of preparation: solubilization, casting, and drying. Firstly, one dispersion/solution of a suitable solvent is prepared and then spread on the surface of the material. Secondly, the prepared is dried since this technique uses the evaporation of the solvent to increase the solution viscosity, which will help to make an edible film with desirable properties [52]. The effectiveness of creating a strong and promising film is conditioned by the processing time, temperature, and drying conditions. As this technique is produced in three phases, the processing time is long and has a long drying time, so it is not practicable for commercial film production [53,54]. This technique has been used in several studies with many different foods, like okra leaves [55], cherry tomato [56], and mangoes [57] and plants like yerba mate [58].

Extrusion is a technique that can be applied in the industry, unlike the previous technique. It begins with the preparation of the film-forming solution and its introduction in the feeding zone. The next step is kneading, where the mix is compressed and the pressure, temperature, and density are increased. The last stage is the heating stage where the solution is extruded through a nozzle at a defined speed, and finally, the films are dried [59]. This technique is more sensitive since a high temperature and unsuitable pH may break the polypeptide chain of the protein conditioning the formation of the edible film [60]. The advantage of this process is that allows for a continuous operation with control of the temperature, size, form, and humidity during the process, providing a more structured film and allowing for a better dispersion of the active compounds [53].

### 2.2. Dipping (Immersion)

The dipping process has a wide application even in the laboratory for being a simple process to operate (Figure 1). First, the coating solution is prepared by dipping the food into the coat-forming solution for a certain time. Then, the coated food is left to dry at room temperature, and the excess solution is removed so as not to become a film with a large layer [61]. This method can be repeated with another solution for making a better coating [59]. This technique has the advantage that it is simple, short, and low-cost. The only downside of this process is that it uses a lot of film-making solutions, taking longer in preparation [60].

### 2.3. Electrospinning

Electrospinning is a process mostly used in edible film packaging [62], Figure 1. This filmmaking process is inexpensive, easy to use, and versatile because it only uses an electric field to stretch a polymeric solution's forming structures with a high specific surface so it is possible to coat the materials; that is, an electrostatic force is used to transform a polymeric solution into nanofibrous films. This process demonstrates a high surface–volume ratio and a small pore size distribution [52]. The only disadvantage is that this process presents a weak interaction between the electrolyte fibers conferring bad mechanical properties on the film [60].

### 2.4. Spraying

Spraying is a method where the coating solution is distributed on the surface of the food pieces from the drops that form in the nozzle of the equipment, being the most used in the food industry (Figure 1). The advantage of this method is that it forms a coating with uniform thickness over the whole surface of the food, the quantity of the solution that is necessary is less, the possibility of contamination is lower, and it can be applied to food with large surfaces [59].

Besides the methods already discussed, there are alternative approaches available for the application of edible coatings. One of these techniques is the pouring method where the coating solution is poured onto the surface that needs to be coated. Another method is the foam method, which is generated and applied to the food as it passes through a cylinder. Subsequently, a brush is used to distribute the foam over the food's surface evenly [63].

## 3. Edible Films and Coating Materials

As it has been mentioned before, edible films are thin layers of edible materials that can be used to coat or wrap food products. They serve as a barrier to moisture, gases, and other external factors, helping to extend shelf life and prevent spoilage of foods. Edible films offer several advantages for food preservation, including reduced microbial growth, decreased moisture loss, improved texture retention, and enhanced visual appeal. They can be applied directly to the food surface or used as a packaging material. However, it is important to note that the specific choice of the edible film depends on the food product, desired properties, and processing conditions. Proper formulation and application techniques are crucial to ensure the effectiveness of edible films in preventing spoilage and

maintaining food quality. In the next section, we revise some of the most used biological materials to make this kind of film. For a more comprehensive reading, in Tables 1–3, we summarize the coating agents used in the food industry and their provenance, applications, types of grapes, and shelf-life after protection.

**Table 1.** Summary of the polysaccharides used in the food industry and their provenance, applications, type of grape, and shelf-life after protection.

| Polysaccharides | Provenance | Applications | Type of Grape | Shelf-Life after Protection | Ref. |
|---|---|---|---|---|---|
| Chitosan | Deacetylating chitin that is found in animal shells, such as crustaceans, insect cuticles, and yeast | Food packaging, nutraceuticals, biotechnology, medicine, pharmaceuticals, textiles, agriculture, cosmetics, and environmental chemistry | Table grapes (*Vitis labrusca* L.) | 12 days at room temperature (25 °C) and 24 days under refrigeration (12 °C) | [64–67] |
| Cellulose | Produced by plants, animals, or bacteria | Food packaging | - | - | [2,68] |
| Methylcellulose | Easily prepared cellulose derivate | Food packaging | - | - | [69] |
| Pectin | Can be found in many fruits and vegetables, such as banana peels, red pomelo, watermelon rinds, sugar beet pulp, sunflower heads, tomato pomace, and carrot pomace, but most commercial pectin is obtained from citrus or apple pomace wastes | The food industry as a preservative in jams, jellies, cakes, and food packaging | Table grape (*Red Globe*) | 35 days of storage at 4 °C | [2,70–73] |
| Carrageenan | Extracted from red seaweed, such as *Eucheuma cottonii*, *Mastocarpus stellatus*, and *Hypnea musciformis* | The food industry, cosmetic, environmental sectors, and health sectors | - | - | [60,74] |
| Alginate | Exists in the cell walls of brown algae | Food, cosmetic, and pharmaceutical industries as emulsifying, gelling agents, stabilizers, thickeners, and additives | Table Grape (*Vitis vinifera* cv. 'Italia') | 9 days of storage at 25 °C | [75,76] |
| Arabic Gum | Produced from the Acacia Senegal tree | Industries such as food, pharmaceuticals, cosmetics, textiles, paper, ink, and printing inks | - | - | [77] |

**Table 1.** *Cont.*

| Polysaccharides | Provenance | Applications | Type of Grape | Shelf-Life after Protection | Ref. |
|---|---|---|---|---|---|
| Xanthan Gum | Exists in nature and is produced by *Xanthomonas campestris* | Tissue engineering, the pharmaceutical industry, and the food industry an additive given its thickener, stabilizer, and emulsifier properties. | Grape (*Vitis vinifera* cv. Pinot Noir) | 21 days under cold storage | [78,79] |
| Tragacanth Gum | Obtained from the dry sap of the species *Astragalus* | Food industry and medicine | - | - | [80,81] |
| Guar Gum | Obtained from the seed endosperm of *Cyamopsis tetragonolobus* | Food packaging, pharmacological, and biomedical | - | - | [82–84] |
| Locust Bean Gum | Obtained by crushing the endosperm seeds from the carob fruit pod | Industries of paper, textile, pharmaceutic, cosmetics, and food products, specifically dairy products such as ice cream, yogurt, and cheese as a thickening agent | - | - | [85] |
| Tara Gum | Obtained from the seed endosperm of the Tara tree | In the pharmaceutical industry as an excipient, the food industry as a thickener and stabilizer, and the biomedical field | - | - | [86,87] |
| Fenugreek Seed Gum | Extracted from seeds of the fenugreek plant | Medicinal, pharmaceutical, and nutraceutical fields | - | - | [88] |
| Basil Gum | Obtained from basil seeds | Food industry | - | - | [89] |
| Shellac Gum | Refined from lac resin that is excreted from insects | Green electronics, 3D printing, stealth technology, intelligent sensors, pharmaceutical and food industries | Grape | 7 days at refrigerated (4 °C) and 7 days at temperature ambient (30 °C) | [23,90,91] |
| Gum Ghatti | An exudate gum from the tree Anogeissus Indifolia | Paper-making industry, wax industry, pharmaceutical industry, and food industry as a thickening agent stabilizer and emulsifier | Table grape (*Vitis vinifera* L. Rishbaba) | 60 days of storage at 0 ± 1 °C and 85% relative humidity | [92,93] |

| Polysaccharides | Provenance | Applications | Type of Grape | Shelf-Life after Protection | Ref. |
|---|---|---|---|---|---|
| Persian Gum | Obtained from the bark of wild almond trees | Food applications as a gelling agent, fat replacer, and film-forming agent | - | - | [94,95] |
| Starch | Is composed of amylose (Water-soluble) and amylopectin (water-insoluble) molecules and is found in several foods, such as cassava, corn, millet, wheat, rice, quinoa, sweet potatoes, peas, and tef | The packaging industry and food industry | *Red Crimson* Grape | 21 days of storage at 7 °C ± 0.2 °C | [2,96,97] |
| Agar | Extracted from certain red seaweed | Biomedical, pharmaceutical fields and food industry as a solidifying agent in various food preparation | White grape | 14 to 21 days of storage at 37 °C | [98–100] |
| Pullulan | Is developed by microbial fermentation | Food packaging, adhesive binders, food coatings, encapsulating agents, fat replaces, thickeners | Table grape (*Vitis vinifera* L.) | 13 days of storage at 25 °C | [90,99,101] |
| Inulin | Is found in several plant species, like *Liliaceae*, *Amaryllidaceae*, *Gramineae*, and *Compositae* | The food industry, pharmaceutical, and biomedical fields | - | - | [98] |
| Konjac Glucomannan | Extracted from the tuber of *Amorphophallus konjac* | The food industry as an additive and thickener | - | - | [102,103] |

### 3.1. Polysaccharides

Polysaccharides are used to make edible films to preserve food, especially fruits and vegetables. The use of these polysaccharides to make coatings is very popular due to their low cost and great availability. Polysaccharide coatings offer barrier properties, such as water and gaseous permeability, and create modified internal atmospheric conditions, resulting in significant preservative effects [55]. Antioxidants and antibacterial characteristics are related in the polysaccharide-based edible covering [2]. Table 1 includes a summary of the polysaccharides used in the food industry and their provenance, applications, types of grapes, and shelf-life after protection.

Chitosan is a natural cationic polysaccharide created by deacetylating the chitin that is found in animal shells, such as crustaceans, insect cuticles, and yeast [64]. In recent years, chitosan has been applied in several foods, like sweet cherries [104], kiwifruit [105] mango [106], litchi [107], grape [65,108,109] apricot [110], banana [111], and papaya [112], among others. Prathibhani et al. [105] studied the effect of an edible coating of chitosan in kiwifruit and showed that this coating has an effective preservation technique, reducing respiration activity and delaying the changes in weight loss and firmness, thereby delaying ripening. Melo et al. [65] showed that, in the preservation of table grapes, a chitosan edible film was effective at maintaining the physicochemical, sensory, and microbiological quali-

ties, presenting an inhibitory effect against pathogenic foodborne bacteria. The chitosan edible film was also responsible for delaying the ripening process of grapes, resulting in decreased weight loss, soluble solids, reduced sugar content, increased moisture retention, and preservation of the titratable acidity values. Chitosan addition to other polymers has also been studied. For example, in the study by Bhatia et al. [113], the mixture of chitosan and other polysaccharides and proteins proved to be a better edible film with promising qualities than the use of chitosan alone. An example of this fact is the work of Wang et al. [114]. These researchers showed that the combination of chitosan and gelatin produced a film coating with excellent physical properties, such as hydrophobicity, color, barrier, and thermal properties. This film prevented external forces (temperature, humidity, corrosion, microorganisms, air, light) from damaging and altering the properties of the food, guaranteeing food hygiene and prolonging its shelf life.

Cellulose is the most widespread polymer in nature and is produced by various sources, such as plants, animals, and bacteria [2]. Bacterial cellulose is the most widely used biopolymer in producing edible films and coatings, with the disadvantage of this polysaccharide being no antimicrobial activity [98]. Strnad et al. [68] showed that the combination of cellulose and chitosan is beneficial to produce the perfect edible film since the cellulose improved the mechanical strength, and the chitosan improved the antibacterial and biocompatible properties of the composites. The cellulose and chitin junction also produces a film that is biocompatible, biodegradable, non-toxic, antimicrobial, presents antioxidant activity, has low gas permeability, and compared with protein cellulose, is less immunogenic and non-hemolytic [115]. In the study by Tapias et al. [116] cellulose was extracted from kombucha that was made with a mix of six different herbal infusions (black and green tea, yerba mate, lavender, oregano, and fennel) and a sucrose addition. The results showed that the yerba mate was an infusion with a high-activity antioxidant, advantageous for the properties of a coating. The coating revealed it can retain natural bioactive substances essential for developing active materials, more specifically, for food packaging. Methylcellulose is an easily prepared cellulose derivative and presents high oxygen permeability and water sensitivity, hindering its widespread usage as not being favorable. To improve these disadvantages, Zhang et al. [69] studied the combination of methylcellulose and curdlan gum created by glycose. The results showed that the mix of these biomaterials can be promising in the industry due to the improvement of a hydrophobic oxygen barrier, water vapor barrier properties, better elongation, and exhibiting great potential for use as edible coatings, capsules, and packaging materials.

Pectin is the most used polysaccharide in food processing [74]. It can be found in many fruits and vegetables, such as banana peels, red pomelo peels, watermelon rinds, sugar beet pulp, sunflower heads, tomato pomace, and carrot pomace. However, most commercial pectin is obtained from citrus or apple pomace wastes [70,71]. Pectin polysaccharide structures contain $\beta - 1, 4$-linked galacturonic acid residues. Extraction of pectin allows for reusing the pomace that is not used in the fruit juice industry. The study by Çavdaroğlua et al. [117] showed that pectin removed from the fig stem may be promising since it has been shown to have a higher mechanical resistance than the pectin removed from the apple, being a hydrocolloid with benefits and characteristics similar to that of pectin pulled from citrus. However, other studies have shown that the creation of a pectin coating is not promising because they are usually brittle, have a poor barrier and mechanical properties, and their moisture resistance limits their applications [118,119]. Various methods have been proposed to solve this problem, such as blending pectin with other polymers and incorporating crosslinkers and filler material [119]. Breceda-Hernandez et al. [72] used a pectin edible coating with lemon essential oil to extend the shelf life of Red Glove grapes, and the results were promising since the edible coating prolonged the grape shelf life by 35 days, prevented fungal decay and moisture loss, and consumer acceptance was positive during the 4 weeks of storage and the same as on day 16 when the coating began to detach, inducing a bad appearance to the grape.

Regarding carrageenan, it is a natural polysaccharide extracted from red seaweed, such as *Eucheuma cottonii*, *Mastocarpus stellatus*, and *Hypnea musciformis* [60]. It is a particularly interesting polymer due to its ability to form thermoreversible gels and viscous solutions becoming elastic and stable if $Ca^{2+}$ is added, which are ideal characteristics for the formation of edible films [74]. In the study by Ismillayli et al. [120], carrageenan was used in combination with chitosan to preserve the vitamin C of the pineapple and reduce weight loss. Zhang et al. [121] created an edible film with a mix of alginate, carrageenan, and shellac gum and tested it on cherries and tomatoes. The treated vegetables showed higher scores in color, odor, pH, and vitamin C content and also could keep the freshness for 7 days. The same edible film was used by the same authors [121] with chicken breast, which showed lower scores in mass loss and color but higher scores in odor and pH and had a better ability to keep than the control samples (chicken breast without film).

Alginate is a polymer that exists in the cell walls of brown algae [75]. Alginate covering materials are created by combining divalent cations, such as $Mg^{2+}$, $Ca^{2+}$, $Al^{2+}$, $Mn^{2+}$, and others, and is used to make strong edible coatings or films with low water resistance due to their hydrophilic aspect [122]. A disadvantage of this polymer is its poor mechanical property; however, according to several studies, the mixture of alginate with pectin and carrageenan improves the missing property and forms a more protective and effective film [115]. In the study by Annisa et al. [123], the mixture of alginate with pectin, gum acacia, and carrageenan prevents degradation, improves thermal and chemical stability, reduces toxicity, increases the effectiveness of active substances, and improves mechanical properties, being an advantageous and beneficial mixture for this use. In the study by Souza et al. [76], an edible coating made with a mixture of alginate (2%), galactomannans (0.5%), cashew gum (0.5%), and gelatin (2.0%) to improve the shelf life of grapes showed weight loss, improved the content of phenolic compounds, and maintained a great physical aspect at nine days of storage.

Gums are naturally occurring polysaccharides that have worldwide industrial uses and have been increasingly investigated for their desirable advantages in creating edible films [123]. Two categories divide gums: exudate gums and seed gums. Seed gums are the best known and used because they are easily accessible since they are extracted from plants, more specifically, from the epidermis of seeds, leaves, and bark. This category includes guar, locust bean, tara, tamarind, basil, and fenugreek gums. The exudate gum is used as a thickener, stabilizer, rheology modifier, soluble fiber, and fat replacer. An example of exudate gum is gum ghatti, Persian, tragacanth, and shellac. Other gums are greatly used, like Arabic gum and xanthan gum [28]. Arabic gum is produced from the Acacia Senegal tree, and it has antioxidant and antifungal characteristics against fungi and antimicrobial properties advantageous for the creation of films. It has already been applied in the edible coatings industry in cereals [77] and fruit conservation [124]. The study by Molnar et al. [125] used the combination of Arabic gum with chitosan, thus improving the properties of chitosan and making an alternative for the creation of edible coatings. Tahir et al. [126] showed that a coating with 15% of Arabic gum was more favorable in extending the storage life of strawberry fruit in cold storage by delaying or maintaining fruit quality and biological properties for 10 days. Moreover, Arabic gum was used in the conservation of fruit, such as banana [127], red raspberry [128], peach [129], pomegranate [130], and guava [131], and showed a favorable coating material that improves the quality and the shelf life of the fruits. Xanthan gum is a degradable and biocompatible polysaccharide that exists in nature and is produced by *Xanthomonas campestris* [78]. Its formation is influenced by temperature, pH, carbon sources, high pressure, polymer concentration, and viscosity. It also presents high solubility in cold and hot water [48]. Lima et al. [132] studied the application of edible films of chitosan and xanthan gum in the conservation of refrigerated fish conservation and showed that this combination presented excellent antimicrobial properties, but compared to other studies, the properties of solubility, the water vapor permeability values, and the loss of mass did not change with the xanthan gum addition. For the preservation of grapes, Golly et al. [79] showed that a coating with

xanthan gum and ascorbic and citric acid is a coating that preserves phytochemicals, color, antioxidant, and texture properties of the grape in cold storage and also extends the shelf life of the grape.

Guar gum is obtained from the seed endosperm of *Cyamopsis tetragonolobus* and according to several studies, has great film-forming capacities, is biocompatible, delays the loss of quality, and prolongs the shelf life of food. The study by Jiang et al. [82] demonstrates several procedures to improve this film: changes to the biopolymer, the addition of plasticizers, mixing with other polymers, the layer-by-layer (LBL) assembly technique, or the addition of plant extracts (Pes) and essential oils (Eos), thus showing improvements in the realization of edible coatings. In the study by Shubham et al. [133] the guar gum (1.5%) edible film created to preserve the quality of litchi was promising because it was capable of forming a protective barrier on the surface of litchi, maintaining fruit total soluble solids, pH, acidity, ascorbic acid, total sugar, reducing sugar, and non-reducing sugar over 12 days. Locust bean gum is a natural polymer and is obtained by crushing the endosperm of the seeds from a carob fruit pod [85]. To create an edible film, it is necessary to put the locust bean gum in hot water (80 °C) for solubilization to occur [98]. In the study by Li et al. [134], strawberries and cherry tomatoes were coated with locust bean gum and cellulose nanocrystals, and the weight loss was lower and showed good mechanical properties, good isolation, a protective effect on the surface, and an excellent moisture and oxygen barrier property that prevented moisture loss and oxygen penetration. Tara gum is a galactomannan polysaccharide of high molecular weight that is obtained from the seed endosperm of the Tara tree [86]. It is soluble in cold water and is a cheaper alternative to locust bean gum and guar gum [86,87]. The disadvantage is that it presents low tensile strength and poor mechanical properties. Carboxymethylation gives the polymer an anionic character, forming a rigid gel with higher viscosity; grafting is used with a polyacrylic and enhances the water absorption characteristics, improves stability, and increases tensile strength; sulfation is used with pyridine and chlorosulfonic acid that reduces flexibility; and crosslinking is used with acrylic acid, and an increase in the amount of the crosslinking agent increases the crosslinking density and provides a more rigid structure, which in turn increases the elastic modulus [87]. Fenugreek seed gum is extracted from seeds of the fenugreek plant and is employed in the medicinal, pharmaceutical, and nutraceutical fields [88]. This gum has the highest solubility and is also soluble in cold water [88,135]. The only disadvantage is that it hydrates slowly because of the absence of an electrostatic change [88]. In the study by Al-Shammari et al. [136], fenugreek seed gum was added to the storage of bread, and the results showed that this gum had lower staling, improved the bread quality, and increased the softness of crumb bread. No result has been found in the bibliography of its application in grapes or its by-products. Basil gum is a hydrocolloid polysaccharide obtained from basil seeds [89,137,138]. Moradi et al. [139] studied the creation of an edible film with basil gum and echinacea extract on the postharvest shelf life of fresh strawberries and showed that the best combination was 3% of each other. This combination showed the lowest growth of microorganisms, weight loss, softening and ascorbic acid, phenol, and anthocyanin degradation and showed the highest antioxidant and superoxide dismutase activity, the lowest peroxidase activity, and received the highest ranked sensory attributes. The increase in the shelf-life of the fresh strawberries in this study was 20 days, which was a great and reassuring result.

Gum ghatti is an exudate gum from the tree *Anogeissus Indifolia* which has good water, oil emulsification stability and is a non-starch polysaccharide. The study by Eshghi et al. [92] that produced an edible coating to improve the quality of a specific type of table grape (Rishbaba) with a mix of chitosan and gum ghatti in different concentrations showed that the addition of gum ghatti to chitosan presented extra beneficial effects for weight loss and the titratable acidity, pH, and total soluble solids of the table grape. The mix of chitosan (1%) and gum ghatti (1%) showed the best concentration in forming an edible coating because it delayed losses in ascorbic acid, presented good membrane stability, and improved antioxidant enzyme activities of the Rishbaba grape during 60 days of storage at

$0 \pm 1$ °C and 85% relative humidity. The film extended the shelf life as well as preserved the bioactive ingredients [92]. Tragacanth gum is an anionic biopolymer obtained from the dry sap of the species *Astragalus* that contains intumesce and water-soluble parts. It has a hydrophilic property that causes stabilizing and emulsifying properties [80,81]. In the study made by Naeini et al. [140], the application of a film with tragacanth gum controlled the reduction in quality, appearance, and deterioration of pomegranate as well as controlled the pH, phenolic content, and antioxidant activity. In another study by Jahanshahi et al. [141], the same coating was applied to tomatoes and decreased weight loss and fungal contamination, increased firmness, and improved appearance, which are advantageous and promising factors. Persian gum is a natural carbohydrate polymer that is obtained from the bark of wild almond trees [94]. Gahruie et al. [142] showed that the mix of gelatin (0.2%, w/v) and Persian gum (0.2% w/v) presented an edible film more mechanical than an edible film only with Persian gum, as the resistant viscosity increased significantly with a slight decrease in transparency and a significant decrease in water barrier properties. This gum has water-soluble and insoluble fractions. Soluble fractions quickly solve in cold water, and the other fraction dissolves partially in hot water. The water-soluble fraction forms brittle films, while the water-insoluble fraction has no film-forming ability. Shellac gum is a natural polymer refined from lac resin that is excreted from insects [23,143]. Although it is insoluble in water, it dissolves in alkaline solutions when cooled due to its acidic nature and has low stability [90]. The addition of shellac gum in the composite films might reduce the water transmittance, increase the mechanical strength, and improve the thermal stability of Konjac glucomannan as shown in the study by Du et al. [143]. The study by Vaishali et al. [91] showed the creation of an edible coating with shellac in the quality of a grape, and the results showed that the titratable acidity decreased in 7 days, but the values remained more constant when the grape was in refrigeration (4 °C). The weight loss was lower when the concentration of shellac was higher (20%). The sensory score was positive for the concentration of 8% and 12% but was classified as overall acceptable.

Starch is a natural polymer composed of amylose (water-soluble) and amylopectin (water-insoluble) molecules that can form strong gels [2]. It is one of the most widely used polymers in the packaging industry and food industries. There are several foods with the presence of starch that can be used for the formation of films, such as cassava, corn, millet, wheat, rice, quinoa, sweet potatoes, peas, and tef [96]. The disadvantage of this polymer, like so many others, is the fact that the mechanical quality is not good [73], and its hydrophilic nature is high, thus making the film permeable to water and sensitive to moisture, leading to reduced film permanence [99]. To improve the mechanical properties of starch-containing films, the study by Tafa et al. [96] showed that it was possible to mix starch with plasticizers and agar to improve the desired properties. To improve the postharvest of refrigerated grapes Fakhouri et al. [97] created an edible coating with starch and gelatin and applied it to the red crimson grape. The results revealed that this combination makes the grape have a good appearance for 21 days as well as increases the mechanical resistance. The good sensorial acceptance by the consumers showed that the utilization of this edible coating does not influence the taste of grapes.

Agar is a polysaccharide extracted from certain red seaweed that possesses excellent film-forming ability and has the particularity of being insoluble in cold water but soluble in boiling water [96]. The application of this polymer is limited due to its shortcomings of strong hydrophilicity and a weak mechanical property, and it is relatively brittle, has low elasticity, and has poor thermal stability [144,145]. To overcome these limitations, the solution is the combination with other polymers, hydrophobic materials, plasticizers, nanoparticles, and antimicrobial agents [144]. Huang et al. [146] used chitosan to improve agar film, and this combination improved the mechanical properties, water resistance, transparency, tensile strength, and elongation. In the study by Kumar et al. [100], an edible coating with agar and zinc oxide nanoparticles to the extension of white grape was created, and the results showed a fresh appearance from 14 to 21 days.

Inulin is a fructan polysaccharide found in several plant species, like *Liliaceae*, *Amaryllidaceae*, *Gramineae*, and *Compositae*, that is soluble in hot water and slightly soluble in cold water. Despite the fact it presents viscous properties, the viscosity decreases with an increase in temperature. In several studies, inulin is considered an alternative option for sugar sweeteners and is used as a stabilizer [98]. In the study by Temiz et al. [147], inulin was added to gelatin film to preserve strawberries, increasing the shelf life of the strawberries, decreasing the weight loss, not changing the quality parameters during storage (20 days), and slowing down the fungal growth.

Konjac glucomannan is a water-soluble natural polysaccharide extracted from the tuber of *Amorphophallus konjac* that has been greatly used as a material of film that exhibits good film-forming ability. The low water resistance and low mechanical properties are disadvantages and limit the use of this polymer [102]. For this, the combination of Konjac glucomannan with other polymers is beneficial. The study by Chen et al. [148] showed the combination of this polysaccharide with curdlan as an excellent edible film to preserve cherry tomatoes once it improved the compatibility, hydrophobicity, and water barrier properties while decreasing weight loss and reducing decay. Firmness, total solid content, and total acid content were maintained. The preservation and the extent of the shelf life of the cherry tomatoes were effective and promising.

Pullulan is a protein that is developed by microbial fermentation and has a starch-like structure. The existing limitations of pullulan-based films are the hydrophilicity, brittleness, high cost, and lack of active functions [149]. Improving this characteristic can be performed with physical and chemical crosslinking, plasticizers, and combination with other polymers, such as pectin [150], chitosan [151–153], starch [154], sodium alginate [155], gelatin [156], konjac glucomannan [102], and others. Yan et al. [102] studied an edible film with a combination of konjac glucomannan and pullulan in the preservation of strawberries. The authors showed that this edible film improves the qualities of this fruit during storage time (14 days) due to its mechanical and barrier properties, decreased weight loss, and maintained skin color. According to this study, the strawberries survived more easily in a low-temperature environment, which is an advantage in the use of films in fruits. In the study by Piña-Barrera et al. [101], an edible coating was created with a combination of pullulan, polymeric nano-capsules, and essential oil of *Thymus vulgaris* to increase the shelf life of table grapes and showed that this coating maintained the grape characteristics of color, firmness, titratable acidity, and total soluble solid content during 13 days of storage at 25 °C.

### 3.2. Proteins

Gelatin comes from animal sub-products, more specifically, pig skin, beef skin, pork, and cattle bones [99]. An edible film based on gelatin proves to be a cheap, biodegradable option and can absorb ultraviolet light due to the aromatic amino acids present in its structure. It has the disadvantage of having low thermal stability and weak mechanical and barrier properties, being more advantageous with the combination of other biopolymers [80].

In the study by Chen et al. [157], cherry tomatoes were coated with an edible film of gelatin, chitosan, nisin, and corn starch and showed that weight loss and total bacterial count were reduced, and higher firmness and better color were observed during the storage over 22 days. The shelf life was more promising than the one presented by the fruit without any coating. In the study by Fatima et al. [158], the researchers created an edible coating with gelatin, chitosan, and zinc oxide to increase the shelf life of fresh grapes, and the results proved this edible coating reduced the browning index and weight reduction of fresh grapes, restricted the microbial growth, and the grapes were attractive at 14 days. Table 2 summarizes the proteins used in the food industry and their provenance, applications, types of grapes, and shelf-life after protection.

**Table 2.** Summary of the proteins used in the food industry and their provenance, applications, type of grape, and shelf-life after protection.

| Proteins | Provenance | Applications | Type of Grape | Shelf-Life after Protection | Ref. |
|---|---|---|---|---|---|
| Gelatin | Animal sub-products, such as pig skin, beef skin, and pork and cattle bones | Confectionery, pharmaceuticals, meat, cosmetics and health products, desserts, and dairy products | Fresh grapes | 14 days at the storage | [66,99,158] |
| Zein | Extracted from corn that is mostly present in corn residues | Food packaging, tissue engineering, food preservation, and cementing the walls of medicines sensitive to microorganisms | - | - | [159–161] |
| Whey Protein | A by-product of cheese manufacturing | Food packaging | *Thompson* grape | 14 days at 25 °C | [162] |

Biodegradable polymers have experienced a notable upswing in popularity as of late, motivated by two main reasons: firstly, the rapid decrease in oil reserves worldwide and second, the urgency to reduce environmental pollution resulting from the use of non-biodegradable resources [163]. Zein, a prolamin polymer that has been commercially available since 1938 [164], is a storage protein extracted from corn that is mostly present in corn residues [73,159]. It has the particularity of not dissolving in water, and the preparation of the film must be made in a mixture of alcohol–water and acetic acid at alkaline pH and high concentrations of urea [160]. In the study by Mouzakitis et al. [165], wheat bread coated with zein-based coatings following storage (4 days, 25 °C) exhibited retardation in moisture migration from crumb to crust compared to their uncoated counterparts. The addition of sunflower oil to this film showed a lower rate and extent of crumb staling compared to uncoated pieces of bread. The sensorial evaluation presents a better crumb texture after 4 days of storage. Xiang et al. [166] studied different edible films with proteins and polysaccharides to preserve meat. The film from proteins, especially zein and glutenin, was the most promising for meat applications because it presented the best results in adhesion, proliferation, differentiation, and mechanical properties.

The promising properties of polylactic acid (PLA) polymers have long been officially recognized for a wide range of advanced applications. In the field of innovation of the surface functionality of final zein products, hydrophobic biopolymers are sought to form double or multilayer films to be water resistant. PLA has excellent mechanical properties, is biodegradable, and is economically attractive. Thus, it becomes an attractive coating material for packaging, artificial blood vessels, tissue engineering scaffolds, and drug carriers [167–169]. According to Chen et al. [170], the presence of a porous PLA coating resulted in significant improvements in several properties of the zein films, such as surface water repellency, the ability to resist fracture, the ability to block the passage of water vapor, and thermal stability.

Whey protein is a valuable by-product of cheese manufacturing that presents a good barrier against aromas, lipids, and oxygen and has good mechanical properties, but its hydrophilic nature limits its capacity to act as a barrier against water vapor [162]. This limitation can be improved with enzymatic modification, like the addition of transglutaminase, and also with the addition of numerous hydrophobic compounds, waxes, fatty acids, or acetylated monoglycerides like vegetable oils [171]. This protein needs the addition of a plasticizing agent to minimize the brittleness and extensibility of the film structure [172]. Glycerol is an example that can be used to improve the characteristics of an edible film with whey protein and is promising due to its ability to create a film with good mechanical and excellent oxygen barrier properties [173]. Other compounds are used for the same effect,

such as jojoba oil, which was used in fresh-cut root parsley [174], Portuguese green tea extract for the preservation of Latin-style fresh cheese [162], ginger and rosemary essential oils to improve the quality of minced lamb meat [175], and others. Dianin et al. [176] studied the creation of an edible coating with whey protein and the *Lactobacillus casei* probiotic for application in tomatoes and grapes and showed that this coating increases the shelf life of grapes but not of tomatoes, and the appearance of grapes was great for 14 days at 25 °C.

### 3.3. Lipids

Different types of waxes can be used in the food industry. Table 3 summarizes the lipids used in the food industry, their provenance, and applications. An example is carnauba wax, which is extracted from the leaves of the carnauba trees and has more stable characteristics; it has excellent water vapor barrier properties, which have been widely recognized, and exhibits a high melting point and hardness, which can be used to develop a durable coating [54]. The study by Lin et al. [31] revealed that the combination of carnauba wax with sodium alginate is not sufficient to create an effective edible film, so added glycerin and calcium ascorbate improves the preservative effect on fresh-cut apples.

**Table 3.** Summary of the lipids used in the food industry, their provenance, and applications.

| Lipids | Provenance | Applications | Ref. |
|---|---|---|---|
| Beeswax | Produced naturally by honeybees in the bees' hive | Production of candles, metal casting, cosmetic products, textiles, polishes, and food processing | [177] |
| Candelilla wax | Obtained from leaves of *E. antisyphilitica Zuccarini* | Food packaging | [178] |
| Carnauba wax | Extracted from the leaves of the carnauba trees | Food packaging, polishing wax, cosmetics, and dentistry | [31,179] |

Beeswax is produced naturally by honeybees in the bees' hive and is well known as hydrophobic in nature and is used in the production of candles, metal casting, cosmetic products, textiles, polishes, and in food processing [177]. It presents high plasticity, is viscoelastic, and has a low melting point [54]. This wax is beneficial in combination with other polymers to make an edible film with good characteristics. For example, the addition of different concentrations (0, 2.5, 5, and 10 wt%) of beeswax in the edible film of cassava starch improved the moisture barrier and reduced the water solubility properties. Nevertheless, it also reduced the tensile strength, elongation, and flexural strength while improving the tensile modulus and flexural modulus until 5 wt% beeswax [177].

Candelilla wax is a plant wax obtained from leaves of *E. anti-syphilitic Zuccarini* and contains the highest number of hydrocarbons compared to other waxes. It is a wax with very low water vapor permeability and is used as an additive and glazing agent in the food industry [178]. Candelilla wax was used in combination with the tarbush plant in the coating of apples, and the results showed that during 8 weeks of storage, this film presented reduced physicochemical changes in the apple. In terms of taste, these films were not promising once the taste and appearance of apples without the edible film was preferred [180].

## 4. Using Essential Oils, Plasticizers, Extracts, and Crosslinker Agents in the Edible Film with Biopolymers

To improve the efficiency of some edible films and coatings, some strategies have been applied. The addition of essential oils is an example that can help in the antibacterial effect [181,182] and retains the qualities and benefits of food. There are several studies with the incorporation of different essential oils, such as thyme [183], oregano [184], grape seed, sea buckthorn [61], lemon [185], ginger [186], rosemary [175], clove [89], cinnamon [187], and lemongrass [130], among others.

Aloe vera is used as a plasticizer that can decrease weight loss and inhibit the maturity stage, as shown in the titratable acidity, pH, and total soluble solids. It has also been studied in several foods, such as tomatoes [25] and apples [185]. The authors showed that aloe vera could minimize the degradation of phenolic and flavonoid compounds and protect the ability of tomatoes to act as an antioxidant, and the application in apples showed an attractive natural shine quite similar to that of freshly harvested fruit.

Propolis extract is a resinous material collected by bees. This extract is used as a natural preservative and has the characteristic of inhibiting the growth of food-contaminating fungi and can replace synthetic fungicides [29]. It was applied in the packaging of meat [188] and rainbow trout fillets [189].

On the other hand, vanillin is a phenolic aldehyde organic compound derived from the vanilla bean that has inhibitory effects against yeasts, molds, and bacteria, thus controlling the decay of the fruit. Vanillin was studied in the conservation of tomato fruit [30] but has few studies with utilization in edible films.

Crosslinker agents are another alternative to improve the quality of edible films and have been studied recently. There are two types of crosslinker agents, green and chemical. Chemical crosslinker agents like, for example, glutaraldehyde have high reactivity, strong crosslinking properties, and high stability, but their toxicity limits their application in food packaging films [190,191]. To resolve this problem, the option of the use of green crosslinker agents (Table 4) has been studied, like, for example, tannic acid, which has excellent antibacterial, antioxidant, and other properties that are referenced in Table 4 that improve the performance of the biopolymers to apply in food packaging films [190,192]. In the study by Zhang et al. [190], tannic acid affects the performance of biopolymer-based food packaging films, like the UV light barrier, the gas barrier, mechanical, antioxidant, and microbial properties, and water sensitivity. In addition, tannic acid crosslinkers significantly improve the effects of biopolymers and have beneficial properties, like citric acid, gallic acid, cinnamaldehyde, ferulic acid, and boric acid.

**Table 4.** Advantages and disadvantages of green and chemical crosslinker agents.

| Crosslinker Agents | | Advantages | Disadvantages | Ref. |
|---|---|---|---|---|
| Green | Tannic acid | • Excellent antibacterial, anti-tumor, antimicrobial, anti-inflammatory, and antioxidant properties;<br>• Low cost;<br>• Excellent biocompatibility;<br>• Strong adhesion properties;<br>• Is widely used in the food industry and is safe;<br>• High solubility in water, ethanol, and acetone;<br>• Non-toxic, renewable, and biodegradable. | • Is not molecular homogenous. | [190,193–195] |

Table 4. *Cont.*

| Crosslinker Agents | | Advantages | Disadvantages | Ref. |
|---|---|---|---|---|
| | Citric acid | • Non-toxic nature;<br>• Antioxidant activity;<br>• pH responsiveness;<br>• Bacterial properties;<br>• Anti-browning effect;<br>• Low cost;<br>• Safe;<br>• Soluble in water;<br>• Acts as a surface barrier;<br>• Increases the flexibility and water solubility of the film;<br>• Has capacity to react and stabilize polysaccharides and materials with high efficiency;<br>• Improves the thermal stability and miscibility of several biopolymers. | • In high concentrations, it becomes a plasticizer. | [196–200] |
| | Gallic acid | • Highly antioxidant, antimicrobial, and antibacterial activity;<br>• Additional health benefits;<br>• Safe;<br>• Effective preservative;<br>• Low cost;<br>• UV-blocking properties;<br>• Oxygen permeability barrier. | - | [198,201] |
| | Cinnamaldehyde | • Highly effective antimicrobial agent. | - | [202] |
| | Ferulic acid | • Antioxidant, antimicrobial, and anti-cancer;<br>• Prevents oxidative stress;<br>• Prevents harmful radiation effects. | • Limited thermal stability;<br>• Cytototoxicity;<br>• High cost and efficiency. | [197,203] |
| | Boric acid | • Functional and safe;<br>• Eco-friendly material;<br>• Antibacterial;<br>• Water soluble;<br>• Commercially viable;<br>• Cheap. | • Cytototoxicity;<br>• High cost and efficiency. | [197,204,205] |
| Chemical | Glutaraldehyde | • High stability;<br>• Low cost;<br>• Improves the hydrophobicity and tensile strength;<br>• Reduces water vapor permeability;<br>• Reduces swelling index;<br>• Enhances rigidity and antibacterial properties. | • Toxic, what limits their application in food packaging films. | [190,191] |
| | Formaldehyde | • Makes food bright in color and stronger toughness and elasticity;<br>• Longer shelf-life;<br>• Reduces bacterial load. | • High toxic;<br>• Carcinogenic. | [206–208] |

**Table 4.** *Cont.*

| Crosslinker Agents | Advantages | Disadvantages | Ref. |
|---|---|---|---|
| Transglutaminase | • Improves thermal stability;<br>• Improves the emulsifying properties;<br>• More chemical crosslinking. | • Are susceptible to deactivation. | [209] |
| Acrylic acid | • It is rich in a large number of active functional groups giving polymer materials excellent properties. | • Inorganic;<br>• Expensive;<br>• Toxic. | [210,211] |

Deep eutectic solvents (DES) were defined as one system composed of a mixture of at least two compounds, a hydrogen-bond acceptor (tetraalkylammonium, quarternary ammonium, or phosphonium salts-HBA) and a hydrogen-bond donor (acids, amines, or alcohols-HBD), which are capable of self-associating [212,213]. It is a safe chemical that recently became a hot topic for use in the manufacturing of biopolymer-based films due to it being inexpensive, presenting a very low environmental impact, having a low toxicity, high renewability, high biodegradability, remarkable tunability, low cost, and the particularity of having outstanding solvation properties that make it perfectly suited for the extraction of bioactive compounds from various natural matrices [41,190]. Only in the last six years was it commonly used as a substitute for expensive organic solvents, and year by year, it attracts more attention [214]. The mixture of choline chloride–citric acid (ChCl-CA), choline chloride–urea (ChCl-Urea), and choline chloride–glycerol (ChCl-Gly) are the DESs most used for plasticizing edible film [215]. For better efficiency and improved quality, natural solvents have been developed that are follow the principles of green chemistry. They are called natural deep eutectic solvents (NADES) and are widely used in the food industry. In the study by Meenu et al., application of an edible film of starch and NADES (lactic acid and fructose) on strawberries showed a delayed ripening process during storage at 20 °C and a reduction in firmness, total soluble solids, weight loss, ascorbic acid, and microbial load.

## 5. Application of Edible Coatings for Preparation of Grapevine By-Product Infusions and in Freshly Processed Products

Natural products are more often searched for in human consumption due to the benefits that can be tapped. However, there is a lack of effective and accessible conservation methods and techniques aimed at reducing postharvest losses. Despite this, one promising technique is the application of edible coatings, which offer a simple, eco-friendly, and effective approach to postharvest preservation. Furthermore, many coatings have the potential to improve the quality and safety of fresh produce, extending shelf life.

The functionality of the edible coating, such as its ability to retain moisture and prevent the transfer of gases, such as water vapor and oxygen, is influenced by the formulation. For example, hydrophilic polymers such as alginates and pectins can be used to improve the coating's ability to retain moisture and prevent food dehydration.

Studies dealing with the development of edible/biodegradable coatings have gained popularity since these products are considered to be ecologically correct, especially when by-products are incorporated, as is the example of the development of chitosan-based coatings with vegetable by-products.

Several examples of the use of products with important properties for health can be found in the most varied studies. For instance, grapes and pomace grapes are rich in diverse compounds that can help with various health issues. Grape seeds contain fats, proteins, carbohydrates, and polyphenols and are rich in phenolic antioxidants, such as phenolic acids, flavonoids, resveratrol, and procyanidins, while the grape skin is rich in anthocyanins [216]. Various studies showed that these compounds have different ways to apply to human health. For example, the proanthocyanidins of grape seeds have anti-

inflammatory, antiapoptotic, antioxidant, and free radical-scavenging properties. The study by Liu et al. [217] showed that they have a neuroprotective effect. Resveratrol has also been found in grapes, and its responsibility for the low incidence of coronary heart disease and for preventing amyotrophic lateral sclerosis, cancer growth, and kidney aging [218] has been shown. Various studies have investigated different ways of reusing these benefits, like the creation of coatings with grape seed extract, grape pomace extract, and grape seed oil, the creation of coatings of grape or grape pomace, and the creation of coatings to grape and grape pomace to make infusions.

The study by Tauferova et al. [21] used grape extract to develop bioactive edible coatings with the junction of chitosan and showed that the extract with 5% of grape pomace has a great classification in sensorial evaluation. Zhao et al. [108] showed the creation of packaging with grape seed extract and indicated that this extract could improve the antimicrobial and antioxidant activities. In another study, they created edible films with only skin pomaces of two different varieties, one red and another white, showing a film with high values of phenolics compounds, more specifically, anthocyanins in the red variety and flavonoids in the white variety. In general, studies showed that both are promising films due to them being raw materials and abundant [219].

Infusions are an example of the reuse of the benefits of several plants and by-products. Ferrer-Gallego and Silva [220] refer that by-products of grapes, especially grape pomace, are used in most flavored waters and infusions due to their high polyphenol content, enzyme activities, antimicrobial effects, antioxidant, and anti-inflammatory activity. Infusions of by-products of grapevine have been mentioned in several studies and described as a promising alternative to reuse grapevine material. In the study by Goulas et al. [221], they used a grape pomace dried in the sun and created an infusion that studied the influence of the ratio of water to grape pomace powder, infusion time, and sensorial evaluation. For the manufacturing of the infusion, the pomace grape was mixed in 200 mL of water heated for 12.2 min at 95 °C, demonstrating the best method to prepare the infusion. This infusion showed antimicrobial effects against some bacteria, such as Listeria monocytogenes serotypes, and the sensory showed to be neutral and flat, making it necessary to add Mediterranean aromatic plants to improve the taste. While the grapes have a reduced time of storage, it is beneficial to find an alternative to preserve the present compounds. In the study by Vilela et al. [109], grape pomace and dried–minced grapes coated with different coating agents were used to produce infusions. The biomaterials used for the coatings were agar–agar, alginate, Arabic gum, chitosan, and gelatin. The results found were important, as it was immediately possible to verify the protective effect of the coatings since the grape encapsulated with the matrices maintained its integrity two months after the coating. Furthermore, when comparing colorimetric parameters, it was observed that the infusions did not show significant color changes except for the chitosan coating. Still, according to this study, alterations tend to occur in the antioxidant capacity, but these cannot be certainly attributed to the coatings. However, some encapsulated agents can change the sensory characteristics of the infusions, although none of the matrices showed negative characteristics in the infusions. It is concluded in this work that the coating of this type of product, in addition to integrating the principles of the circular economy, provides the consumer with an innovative way of preparing a grape infusion, preserving the nutraceutical and sensory characteristics. So, new studies should now focus on the effects of the coatings on grape and pomace infusions with an interest in studying the best combinations of biopolymers that can retain the benefits of grapes for a maximum time and verify the integrity of the coatings while undergoing dissolution in water. Since there are not many studies that showed the potential of edible films to create infusions of grapes, it is interesting to deepen our knowledge on this matter due to the benefits that the grape's reuse can have on the improvement of human health. The infusion can be studied in terms of the physicochemical parameters (°Brix, pH, phenolic compounds, antioxidant activity, colorimetric parameters) as well the evaluation of the consumer with sensorial analysis to understand the acceptance of the consumer.

Furthermore, edible films complemented or used with various substances can add flavor, aroma, and functional properties to the films themselves or the foods/infusion material they are used to wrap or coat [222]. Edible films can be infused with natural extracts, essential oils, or artificial flavors to impart specific tastes to the films. For example, mint, citrus, vanilla, or coffee flavors can be infused to enhance the sensory experience of the food. Adding antimicrobial agents to edible films can help inhibit the growth of spoilage-causing microorganisms and extend the shelf life of foods. Natural antimicrobial substances, like plant extracts (e.g., oregano, thyme) or essential oils (e.g., cinnamon, clove), can be infused into the films [223].

Edible films can also be added with vitamins, minerals, or other bioactive compounds to enhance the nutritional value of foods. For example, films can be enriched with vitamins A, C, or E, omega-3 fatty acids, or plant extracts with antioxidant properties [224], allowing the preparation of more nutritional infusions. And, if natural colorants such as fruit or vegetable extracts can be infused into edible films, it is also possible to provide the final product with attractive visual effects, among other properties [225].

Infusing edible films requires careful consideration of the compatibility and stability of the infused substances with the film matrix. Proper techniques, such as solvent-based extraction or emulsification, are used to incorporate the infusions uniformly. The concentration and application method of the infused edible films should be optimized to achieve the desired sensory and functional characteristics while ensuring food safety and stability.

In addition to the application of coatings on vine by-products, an increasing number of products are currently being subjected to coating processes. Maintaining the quality of minimally processed products is a significant challenge due to the difficulty of preserving them for long periods [226]. After harvesting fresh produce, several metabolic and physiological reactions occur. This is especially observed in some fruits and vegetables, which tend to deteriorate due to continuous respiration, perspiration, ethylene production, and ripening. In addition, due to the presence of more phenolic compounds, such as gallic acid, catechin, and caffeic acid, they tend to darken quickly. However, through innovative coating techniques, it is possible to delay these processes and increase the durability of the products, resulting in greater quality and safety until consumption [227]. This way, the use of chitosan nano-emulsion in grapes reduces the initial growth of S. typhimurium, yeasts, and molds and shows retention of the antioxidant capacity [228]. Fuji apples, when sprayed with a combination of 40% aloe vera gel and 1% lemon essential oil hydroxypropyl methylcellulose HPMC 0.1% v/p, delay the ripening process with minimal color change observed [185]. Fresh-cut apple coatings based on edible sodium alginate nano-emulsion showed greater inactivation of *E. coli* and slower growth of psychrophilic bacteria [229]. Strawberries with an edible coating based on *Lippia sidoides* essential oil showed a huge decrease in fungal activity [230]. In bananas, an increase in total soluble sugars and a reduction in the respiration rate during storage were observed when a fibroin-based coating was applied [231]. In another study with banana, Thakur et al. [232] concluded that by coating it with rice starch mixed with sucrose esters, ethylene biosynthesis and respiration rate were reduced and chlorophyll degradation was delayed, extending the shelf life of the fruit by 12 days of shelf life. The application of a coating with ascorbic acid in modified atmosphere packaging on pomegranate arils improved their sensory characteristics, increased shelf life, and maintained the red color of the arils [233]. A plum coating with 3% alginate reduced ethylene production and increasing storage time by 2 weeks [234]. According to Dorostkar and Moradinezhad [235], coating pomegranate with 1% sodium bicarbonate had a beneficial effect in maintaining total soluble solids and titratable acidity and reduced its decomposition. In guava, chitosan coating delayed ripening and improved its antioxidant properties [236]. In green chilies, the same storage quality was observed when coated with shellac and starch, EDTA, and sodium alginate compared to uncoated green chilies [237]. Studies carried out by Massilia et al. [238] with a coating based on alginate with essential oils of lemongrass, palmarosa, cinnamon, and malic acid on melon/fresh showed that the parameters of quality and microbiological safety were maintained with a significant

reduction in the population of Salmonella enteritidis. In the case of mushrooms/fresh, the chitosan-based coating contributed to the reduction of enzymatic activities that lead to decomposition [239].

## 6. Conclusions

Applying edible coatings on fresh or cut fruits and vegetables, as well as on packaging, is a safe and environmentally friendly way to preserve them after they are harvested. These coatings are made from natural, non-toxic materials if consumed and are eco-friendly. This preservation technique is also cost-effective and has shown positive outcomes. By using edible coatings on minimally processed products, we can maintain their nutritional value, keep their color vibrant, preserve their taste and texture, and provide microbial protection.

Edible films based on biopolymers have shown their benefits and are a promising substitution for plastic packaging. Sometimes, a single-component edible film can be used; other times, due to certain weaknesses, it is more practical to use a film made with a combination of several agents. So, usually, it is necessary to use two or more natural substances as coating agents to improve both components and their strengths and allow them to complement each other to obtain a better-performing edible film compared to a single-component edible film.

Current research has shown that foods produced with grape by-products or their extracts represent enormous promise in the prevention of chronic diseases in addition to making the wine and vineyard sector more sustainable. Nowadays, scientists, politicians, winemakers, and the scientific community look for more profitable and sustainable options using wine by-products. To produce an infusion of grape and sub-products of grape, several biomaterials may be used for coatings, such as chitosan, cellulose, pectin, carrageenan, alginate, different types of gums, starch, agar, inulin, konjac glucomannan, gelatin, zein pullulan, whey protein, and different types of waxes. To improve the efficacy of these biopolymers, the combination of matrices is advantageous. For example, the combination of a polysaccharide and a protein improves the characteristics of an edible film; the same occurs for the combinations of chitosan and gelatin, starch and gelatin, chitosan and zein, pullulan and pectin, and tragacanth gum and whey protein. Many combinations can be efficient in the production of an edible film. However, it is imperative to know the properties of the material to be coated, the objectives of the coating, and also the properties of the coating agents themselves or the biopolymer combination so that the choice of coating is the most advantageous. There is an effective lack of knowledge that needs to be suppressed regarding the impact of active components on the functionality and mechanical and sensory characteristics of edible coatings.

For future research, it would be interesting to conduct more research on the application of biopolymer combinations and explore the theme of green crosslinker agents, especially NADES, as they are increasingly becoming promising in the packaging sector and are a safe and healthy option. Due to the beneficial properties of grapes and by-products of grapes, there needs to be more research in the preparation of coatings with different combinations and new biopolymers for the production of infusions since there is not enough recent and large-scale research. For example, the biopolymers cellulose, methylcellulose, carrageenan, Arabic gum, tragacanth gum, guar gum, locust bean gum, tara gum, fenugreek seed gum, basil gum, Persian gum, inulin, zein, beeswax, candelilla wax, and carnauba wax have not had any research conducted in the application of these products, and it will be interesting to perform research on this topic.

Therefore, in-depth and comprehensive research is needed to explore the functionality, formulation, sensory, and mechanical properties of the active components, as well as the application of the coatings, to make them technologically viable and readily available in the market. Though, we must never forget that the product found, no matter how many benefits it brings to health and how sustainable it is, must be attractive to the consumer, hence the importance of always associating biochemical analysis and nutraceutical potential with sensory analysis studies.

**Author Contributions:** Conceptualization, T.P. and A.V.; writing—original draft preparation, T.P., A.P. and A.V.; writing—review, and editing, T.P., A.P. and A.V. supervision, T.P. and A.V. All authors have read and agreed to the published version of this manuscript.

**Funding:** This research was funded by the National Funds from FCT Portuguese Foundation for Science and Technology, Portugal, and COMPETE under the projects UIDB/00616/2020, UIDP/00616/2020, and UIDB/04033/2020.

**Institutional Review Board Statement:** Not applicable.

**Informed Consent Statement:** Not applicable.

**Data Availability Statement:** Not applicable.

**Acknowledgments:** The authors would like to thank the Chemistry Center (CQ-VR) and CITAB-Inov4Agro for their financial support.

**Conflicts of Interest:** The authors declare no conflict of interest.

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
