# Peer review of "Edible Coatings and Films for Preparation of Grapevine By-Product Infusions and in Freshly Processed Products"

_coatings, doi:10.3390/coatings13081350_

Round 1
Reviewer 1 Report (Previous Reviewer 1)
Dear authors, I am sorry, but revised manuscript is not satisfactory. Instead of giving something new, you made a mix of different subjects. My recomendation now is to reject the manuscript.
Minor editing of English language required
Author Response
Reviewer1
Dear authors, I am sorry, but revised manuscript is not satisfactory. Instead of giving something new, you made a mix of different subjects. My recomendation now is to reject the manuscript.
Authors: The authors are grateful for the reviewer work. A review article is a type of academic publication that synthesizes and evaluates previous studies already published, that is, it aims to provide a critical and systematic analysis of existing knowledge on a given topic. This type of manuscript is an important way of consolidating and updating knowledge in a given area, involving exhaustive research of the scientific literature available on the subject in question. The articles are relevant for researchers, students and professionals interested in updating themselves on a given topic, by offering an overview of existing knowledge, highlighting areas of controversy and identifying future research directions.
About the manuscript that we are now presenting, we do not doubt that the topic is relevant. The increasing demand for fresh products is a result of substantial changes in the lifestyle of populations and greater awareness of the nutritional aspects of dietary patterns. The literature mentions that in developed countries post-harvest losses can reach 50%, and inadequate storage can cause serious problems of quality and food safety. In this context, the edible coating of fresh produce appears to be an effective approach to mitigating produce safety and quality issues.
This review explores various types of edible coatings with their impact on the quality attributes of fresh produce, as well as the benefits and key functions of each type of coating. This valuable information can help processors select the appropriate coating material for many fresh and minimally processed foods. Several changes have been made since the last manuscript. Thus, the abstract has been improved and is now more focused on the topic. The introduction was improved in several aspects, namely with regard to trends in the use of chitosan as a primary biopolymer for functional films and the manufacture of coatings for food and natural products; studies on new concepts in film formulations for packaging and the manufacture of dextran/chitosan mixture films for biopackaging were added, and new discoveries and approaches were added regarding Zein, PLA, among other biopolymers. Figure 1 has been deleted and replaced by a schematic illustration of the techniques of making edible coatings and how the coatings protect the grape-vine by-products. Table 1 was separated into several tables based on polysaccharides, proteins, and lipids. So, Table 1 summarizes the polysaccharides used in the food industry, their provenance, applications, type of grape, and shelf-life after protection. Table 2 summarizes the proteins used in the food industry, their provenance, applications, type of grape, and shelf-life after protection, and Table 3 summarizes the lipids used in the food industry, their provenance and applications.
Table 4 containing information regarding the advantages and disadvantages of green and chemical crosslinker agents was constructed. Another important approach added to this work was the role of tannic acid during the formulation of smart films. An equally important topic was also envisaged regarding the new generation of green additives to improve the properties of chitosan films, and food packaging, as Deep eutectic solvents (DES) are safe chemical substances and have recently been used for the preparation of selective films. All information added was based on very current bibliographical references. Conclusions were also improved.
We hope that all these changes met the reviewer's requirements, to whom we again thank the opinion, as it gave us the opportunity to greatly improve our work.
Reviewer 2 Report (New Reviewer)
The authors tried to summarize the application of edible coatings and films on the preservation of grape-vine by-products and they have summarized the techniques and materials for edible film making. However, the review can be improved before publication.
1. Abstract: Lines 13-24 are all about the background of the wine industry, which can be shortened. The abstract should focus on what's new that has been summarized by the authors.
2. Figure 1 is not essential in the review. The authors should add a schematic illustration of the techniques of making edible coatings and how the coatings protect the grape-vine by-products. Just by these words, we can't imagine how edible films protect the grape-vine by-products and what kind of by-products.
3. Table 1 should be specified or separated into several tables based on polysaccharides, proteins, lipids, et al. Details, such as what kind of grape-vine by-product and shelf life after protection, should be added.
4. I suggest the authors should add a figure to illustrate the application of edible coatings in the according part.
5. The conclusions didn't provide an effective summary and hints for the readers. After referring to so much literature, the authors should summarize their own points that can be referred to by later researchers.
Author Response
Please see the attachment.

Reviewer 3 Report (New Reviewer)
The paper entitled "Edible coatings and films for preparation of grape-vine by-products infusions and in freshly processed products", is an interesting topic to review since more safe substances need to be used in food-related items. The reviewer suggests several comments after carefully reading this paper. Please, follow my suggestions to improve the quality of the
paper to be potentially published in Coatings, the comments are given as follows (the reviewer is willing to review the revised version of this paper):
1) Your introduction section needs to be improved in terms of:
1) updated reviews and concepts for edible films.
2) Also, give feedback on Trends in chitosan as a primary biopolymer for functional films and coatings manufacture for food and natural products
3) additional and current studies on new concepts in film formulations for food packaging should be added, Dextran/chitosan blend film fabrication for bio‐packaging.
2) The authors should add a table comparing the advantages and disadvantages of possible green crosslinker agents (tannic acid) used in film preparation compared with chemical crosslinking agents (e.g. glutaraldehyde).
3) The authors need to address the role of tannic acid during the formulation of smart films. Important point.
4) a couple of figures or even a table describing the different methods and strategies for film preparation should be added, e.g. dense film casting methods/ solvent evaporation, etc etc.
5) Review the Different strategies to reinforce the milk protein-based packaging composites? Please take a look and give references in the revision.
6) Zein, PLA, among other biopolymers have not been reviewed in this paper. new findings and approaches need to be added.
7) As a perspective in this field, the authors need to provide a new generation of green additives for improving the properties of chitosan films, Industrial and Engineering Chemistry Research 2022, 61 (48), 17397-17422 This a very hot topic in the case of food packaging since DES are safe chemical substances and have been used recently used for the preparation of selective films.
Few grammar mistakes can be identified in the paper.
Round 2
Reviewer 2 Report (New Reviewer)
The revised version is acceptable
Reviewer 3 Report (New Reviewer)
the paper is ready for acceptance.
This manuscript is a resubmission of an earlier submission. The following is a list of the peer review reports and author responses from that submission.
Round 1
Reviewer 1 Report
Dear Authors,
I am sorry, but my opinion is that submited manuscript has none or very moderate novelty. Reviews of edible food coatings are already available and I have not found nothing new in the manuscript. Title and abstract are misleading because only last subtitle deals with the narrow subject of the manuscript.
Author Response
Dear Authors,
I am sorry, but my opinion is that submited manuscript has none or very moderate novelty. Reviews of edible food coatings are already available and I have not found nothing new in the manuscript. Title and abstract are misleading because only last subtitle deals with the narrow subject of the manuscript.
Authors: the authors are grateful for the reviewer's comments as this is an opportunity to improve our work. We understand the concern of the reviewer and we have modified the document. Thus, the title was modified and the abstract was considerably improved. In addition point 2 was separated into two different points, according to the suggestion of another reviewer. So, point 2 was entitled “2. Edible films and Coatings” and point 3: “3. Techniques of making edible coatings”. In the development of new point 4: “4. Food Matrices”, this was enriched with several current bibliographical references. We have considerably developed now point 6: “6. Application of edible coatings for preparation of grape-vine by-product infusions”. Conclusions were also improved.
In this way, having had a huge improvement in the manuscript, we hope to have met the reviewer's concerns.
Reviewer 2 Report
Dear authors,
in general it is a good review but according to the title and proposed topic. it needs to be more detailed about the grape-vine by-products for edible coatings application.
Section 5. Application of edible coatings for preparation of grape-vine by-products infusions, which is very important according to title, has to be a bit detailed not only treated as general one. Grape pomace, besides other grape-vine by-products sources, is a valuable source of bioactives which are perfect candidate for edible coatings, please mention some of them.
(e.g. please check: https://www.mdpi.com/2073-4360/13/15/2578)
The abstract needs to be improved, it is very poor and does not catch the attention of readers.
Author Response
D
Dear authors,
in general it is a good review but according to the title and proposed topic. it needs to be more detailed about the grape-vine by-products for edible coatings application.
Section 5. Application of edible coatings for preparation of grape-vine by-products infusions, which is very important according to title, has to be a bit detailed not only treated as general one. Grape pomace, besides other grape-vine by-products sources, is a valuable source of bioactives which are perfect candidate for edible coatings, please mention some of them.
(e.g. please check: https://www.mdpi.com/2073-4360/13/15/2578)
Authors: We agree with all of the suggestions and thank you for the article. We add to this topic the next text to the first paragraph, lines 777-790: “Grapes and pomace grapes are rich in diverse compounds that can help with various health issues. Grape seeds contain fats, proteins, carbohydrates, and polyphenols and are rich in phenolic antioxidants such as phenolic acids, flavonoids, resveratrol, and procyanidins while the grape skin is rich in anthocyanins [193]. Various studies showed that these compounds have different ways to apply to human health. For example, proanthocyanidins of grape seed have anti-inflammatory, antiapoptotic, antioxidant, and free-radical-scavenging properties and the study of Liu et al. [194] showed that have a neuroprotective effect. Resveratrol has also been in the grape and is demonstrated that is responsible for the low incidence of coronary heart disease and for preventing amyotrophic lateral sclerosis, cancer growth, and kidney aging [195]. Various studies are investigated different ways of reusing these benefits like the creation of coatings with grape seed extract, grape pomace extract, and grape seed oil, the creation of coatings of grape or grape pomace, and the creation of coatings to grape and grape pomace to making infusions.”
We added a second paragraph, lines 791-799: “The study of Tauferova et al. [20] used a grape extract to develop bioactive edible coatings with the junction of chitosan and showed that the extract with 5% of grape pomace has a great classification in sensorial evaluation. Zhao et al. [58] showed the creation of packaging with grape seed extract and indicated that this extract could improve the antimicrobial and antioxidant activities. In another study, they created edible films with only skin pomaces of two different varieties, one red and another white showing to be a film with high values of phenolics compounds, more specifically anthocyanins in the red variety and flavonoids in the white variety. In general, showed that both are promising films due to their being raw materials and abundant [196].”
In lines 805- 813, we added: “In the study of Goulas et al. [198] they used a grape pomace dried in sun and create an infusion that study the influence of the ratio of water to grape pomace powder, infusion time, and sensorial evaluation. For the manufacture of the infusion, the pomace grape was mixed in 200mL of water heated for 12,2 minutes at 95°C, demonstrating the best method to prepare the infusion. This infusion showed antimicrobial effects against some bacteria such as Listeria monocytogenes serotypes and sensory showed to be neutral and flat being necessary to add Mediterranean aromatic plants to improve the taste. While the grapes have a reduced time of storage its beneficial to find an alternative to preserve the present compounds.”
The abstract needs to be improved, it is very poor and does not catch the attention of readers.
Authors: We appreciate the comments as this is an opportunity to improve their work. the abstract has been improved as recommended by the reviewer: “The wine industry is responsible for a considerable part of the environmental problems because of the large amounts of residues. However, several studies have shown these wine industry residues, such as grapes, skins, seeds, and leaves are a rich source of nutraceutical compounds and present a great potential for the development of new food products. Effectively they represent a complex matrix of bio-compounds, such as phenolic compounds, flavonoids, procyanidins, anthocyanins, tannins, catechin, quercetin, kaempferol, transresveratrol and nutrients such as vitamin C. Current studies have been stating that foods produced with wine and vine by-products or their extracts have antioxidant, anti-inflammatory, cardioprotective and anti-aging anti-cancer activities, etc. which are beneficial to human health. The increasing preference of consumers for natural food additives encourages the use of wine products as an alternative source of natural antioxidants in the food industry. However, due to processing (drying, mincing), some of the vine by-products are perishable and may present a short shelf-life. The protection of the developed products can be achieved through the utilization of edible films and coatings. More research needs to be carried out to optimize coating formulations to achieve the highest possible quality. This review aims to elucidate the different types of edible coatings that can be used in the preparation of grape by-products for foods and drinks, namely grape vine infusions made with dried minced grapes, dried minced grape pomaces, and leaves. Besides the usually used coating material such as chitosan, agar-agar, gelatin, and alginate, other compounds will also be discussed, namely guar-gum, soy lecithin, maltodextrin, inulin, and propolis.”
Reviewer 3 Report
This review aims to elucidate the different types of edible coatings that can be used in the preparation of grape by-products for foods and drinks. However, the logic of organization is poor and the criticism on the cited findings and results is pale. Some suggestions are as follows:
1. Title is “edible coatings”, while in the text is “edible coatings and films”.
2. The reviews associated with the similar topics should be analyzed to enlighten the novelty of this review.
3. Fig.1, the classification is wrong.
4. Are the contents of 2.1-2.4 under the catalogue of 2.
5. Application of edible coatings for preparation of grape-vine by-products infusions needs enhancing.
Author Response
This review aims to elucidate the different types of edible coatings that can be used in the preparation of grape by-products for foods and drinks. However, the logic of organization is poor and the criticism on the cited findings and results is pale. Some suggestions are as follows:
1.Title is “edible coatings”, while in the text is “edible coatings and films”.
Authors: Thank you for the suggestion. The title was modified with the addition of ‘and films’.
2.The reviews associated with the similar topics should be analyzed to enlighten the novelty of this review.
Authors: Thank you for the suggestion, which we agreed on. In this way, we added to the manuscript some current studies where biopolymers were added to grapes or grape by-products and verified their action. So, in line 261 we added: “In the study of Tapias et al. [69] cellulose was extracted from kombucha that was made with a mix of six different herbal infusions (black and green tea, yerba mate, lavender, oregano, and fennel) and the sucrose addition. The results showed that the yerba mate was the infusion that has a high activity antioxidant being advantageous for the properties of a coating. The coating revealed that can retain natural bioactive substances essential for developing active materials more specifically for food packaging.”
In line 310 we added: “In the study of Breceda-Hernandez et al. [81] they used a pectin edible coating with lemon essential oil to extend the shelf life of Red Glove grapes and the results were promising because the edible coating prolonged the grape shelf life by 35 days, prevented fungal decay, moisture loss and the consumer acceptance was positive during four weeks of storage same as on the day sixteen the coating began to detach inducing a bad appearance to the grape.”
In line 363 we added: “In the study of Souza et al. [91] an edible coating made with a mix of alginate (2%), galactomannans (0.5%), cashew gum (0.5%), and gelatin (2.0%) to improve the shelf life of grape and showed a weight loss, improved the content of phenolic compounds and maintained a great physical aspect at 9 days of storage.”
In line 417 we added: “For the preservation of grapes, Golly et al. [107] showed that a coating with guar gum and ascorbic and citric acid is a coating that preserves phytochemicals, color, antioxidant, and texture properties of the grape in cold storage as also extending the shelf life of the grape.”
In line 481 we added a study because we’re the only ones that don´t have an example: “In the study of Al-Shammari et al. [123] fenugreek seed gum was added to the storage of bread and the results showed that this gum had lower staling, improved the bread quality, and increase the softness of crumb bread.” No result has been found in the bibliography of its application in grapes or in its by-products.
In line 542 we added: “The study of Vaishali et al. [137] showed the creation of an edible coating with shellac in the quality of the grape and the results showed that the titratable acidity decreased in 7 days but the values remain more constant when the grape is in refrigeration (4°C). The weight loss was lower when the concentration of shellac was higher (20%). The sensory score was positive for the concentration of 8% and 12% but was classified as overall acceptable.”
In line 568 we added: “To improve the postharvest of refrigerated Red Crimson grape, in the study of Fakhouri et al. [141] they created an edible coating with starch and gelatin and the results showed that this combination make the grape with good appearance during 21 days and increased thickness, mechanical resistance, and sensorily was accepted and the utilization of this edible coating does not influence the taste of grape.”
In line 592 we added: “In the study of Kumar et al. [146] edible coating with agar and zinc oxide nanoparticles to the extension of green grape was created and results showed a fresh appearance from 14 to 21 days.”
In line 646 we added: “In the study of Fatima et al. [155] they created an edible coating with gelatin, chitosan, and zinc oxide to increase the shelf life of fresh grapes and the results showed that this edible coating reduced the browning index and weight reduction of fresh grapes, restrict the microbial growth and the grape were attractive at 14 days.”
In line 696 we added: “In the study of Piña-Barrera et al. [171] an edible coating was created with the combination of pullulan, polymeric nanocapsules, and essential oil of Thymus vulgaris to increase the shelf life of table grapes and showed that this coating maintained the grape characteristics of color, firmness, titratable acidity, and total soluble solid content during 13 days of storage at 25°C.”
In line 720 we added: “Dianin et al. [177] studied the creation of an edible coating with whey protein and Lactobacillus casei probiotic for the application in tomatoes and grapes and showed that this coating increases the shelf life of grapes but not of tomatoes and the appearance of grapes was great for 14 days at 25°C.”
- Fig.1, the classification is wrong.
Authors: We agree, and the subtitle was replaced with ‘edible coatings’.
- Are the contents of 2.1-2.4 under the catalogue of 2.
Authors: Thank you for the correction, so we create a new heading with the name: “3. Techniques of making edible coatings”, changing the numbering of the following headings.
- Application of edible coatings for preparation of grape-vine by-products infusions needs enhancing.
Authors: Thank you for the comments and recommendations. This point has been improved, please see lines 777-799: “Grapes and pomace grapes are rich in diverse compounds that can help with various health issues. Grape seeds contain fats, proteins, carbohydrates, and polyphenols and are rich in phenolic antioxidants such as phenolic acids, flavonoids, resveratrol, and procyanidins while the grape skin is rich in anthocyanins [193]. Various studies showed that these compounds have different ways to apply to human health. For ex-ample, proanthocyanidins of grape seed have anti-inflammatory, antiapoptotic, antiox-idant, and free-radical-scavenging properties and the study of Liu et al. [194] showed that have a neuroprotective effect. Resveratrol has also been in the grape and is demonstrated that is responsible for the low incidence of coronary heart disease and for preventing amyotrophic lateral sclerosis, cancer growth, and kidney aging [195]. Various studies are investigated different ways of reusing these benefits like the crea-tion of coatings with grape seed extract, grape pomace extract, and grape seed oil, the creation of coatings of grape or grape pomace, and the creation of coatings to grape and grape pomace to making infusions.
The study of Tauferova et al. [20] used a grape extract to develop bioactive edible coatings with the junction of chitosan and showed that the extract with 5% of grape pomace has a great classification in sensorial evaluation. Zhao et al. [58] showed the creation of packaging with grape seed extract and indicated that this extract could im-prove the antimicrobial and antioxidant activities. In another study, they created edible films with only skin pomaces of two different varieties, one red and another white showing to be a film with high values of phenolics compounds, more specifically an-thocyanins in the red variety and flavonoids in the white variety. In general, showed that both are promising films due to their being raw materials and abundant [196].”
Lines 805-813: “In the study of Goulas et al. [198] they used a grape pomace dried in sun and create an infusion that study the influence of the ratio of water to grape pomace powder, infu-sion time, and sensorial evaluation. For the manufacture of the infusion, the pomace grape was mixed in 200mL of water heated for 12,2 minutes at 95°C, demonstrating the best method to prepare the infusion. This infusion showed antimicrobial effects against some bacteria such as Listeria monocytogenes serotypes and sensory showed to be neutral and flat being necessary to add Mediterranean aromatic plants to improve the taste. While the grapes have a reduced time of storage it's beneficial to find an al-ternative to preserve the present compounds.”
Lines 828-836: “… being interesting to study what is the best combination of biopolymers that can retain the benefits of grape for a maximum time and verified the integrity of the coatings while the dissolution in the water. Since there are not many studies that showed the potential of edible films to create infusions of grapes is interesting to study deeper due to the benefits that the grape has and can be reused to improve health human. The in-fusion can be studied in terms of physicochemical parameters (°Brix, pH, phenolic compounds, antioxidant activity, colorimetric parameters) as well the evaluation of the consumer with sensorial analysis to understand the acceptance of the consumer.”
Reviewer 4 Report
Line 57-58 The sentence '. Due to its low cost and because it is a resistant and easy-to-mold material its use is high' should be rearrangement. The words 'due to' and 'because' are repeated.
Line 391- What is 'de application' mean?
Line 491- What is 'de mix of gelatin' mean?
Line 702- Comma should be added before 'some'.
Line 746-766 Suggestions and prospects for future scientific research can be added appropriately.
Author Response
Line 57-58 The sentence '. Due to its low cost and because it is a resistant and easy-to-mold material its use is high' should be rearrangement. The words 'due to' and 'because' are repeated.
Authors: We agree with the correction and the sentence was changed to: “Its use is high due to its low cost, resistance, and easy-to-mold”.
Line 391- What is 'de application' mean?
Authors: It´s a spelling mistake, it's been corrected.
Line 491- What is 'de mix of gelatin' mean?
Authors: It´s a mistake as well, it's been corrected to: ‘the mix of gelatin’.
Line 702- Comma should be added before 'some'.
Authors: We agree, and a comma was added.
Line 746-766 Suggestions and prospects for future scientific research can be added appropriately.
Authors: Thank you for the recommendations. The conclusions were improved, pointing out suggestions and perspectives for future scientific research: “Edible films based on biopolymers have shown their benefits and are a promising substitution for plastic packaging. Sometimes, a single-component edible film can be used, other times, due to certain weaknesses, is more practical to use a film made with a combination of several agents. So, usually, it is necessary to use two or more natural substances as coating agents, such that both components improve their strengths and complement each other to obtain a better-performing edible film compared with a sin-gle-component edible film.
Current research has shown that foods produced with grape by-products or their extracts represent enormous promise in the prevention of chronic diseases, in addition to making the wine and vineyard sector more sustainable. Nowadays, scientists, politicians, winemakers and the scientific community look for more profitable and sustainable options using wine by-products. To produce an infusion of grape and sub-products of grape several biomaterials may be used for coatings, such as chitosan, cellulose, pectin, carrageenan, alginate, different types of gums, starch, agar, inulin, konjac glucomannan, gelatin, zein pullulan, whey protein, and different types of waxes. To improve the efficacy of these biopolymers the combination of matrices is advantageous. For example, the combination of a polysaccharide and a protein improves the characteristics of the edible film, the same occurs for the combinations of chitosan and gelatin, starch and gelatin, chitosan and zein, pullulan, and pectin, tragacanth gum and whey protein. Many combinations can be efficient in the production of an edible film. However, it is imperative to know the properties of the material to be coated, the objectives of the coating, and also the properties of the coating agents themselves, or biopolymers combination, so that the choice of coating is the most advantageous. Furthermore, as there are few studies showing the potential of edible films to create grape infusions and grape by-products, it is also important to have a relatively in-depth knowledge of their maximum nutraceutical potential in food applications such as infusions. Though, we must never forget that the product found, no matter how many benefits it brings to health and is sustainable, must be attractive to the consumer, hence the importance of always associating biochemical analysis, and nutraceutical potential with sensory analysis studies.
Round 2
Reviewer 1 Report
Dear authors,
although section 6 was improved in the revised manuscript, my recomendation remains the same. Still, this section might be a good starting point to elaborate further the topic and answer more closely to the Manuscript title.
Reviewer 3 Report
The revision was improved.